# Recent Advancement and Structural Engineering in Transition Metal Dichalcogenides for Alkali Metal Ions Batteries

**DOI:** 10.3390/ma16072559

**Published:** 2023-03-23

**Authors:** Nabi Ullah, Dariusz Guziejewski, Aihua Yuan, Sayyar Ali Shah

**Affiliations:** 1Department of Inorganic and Analytical Chemistry, Faculty of Chemistry, University of Lodz, Tamka 12, 90-403 Lodz, Poland; 2School of Environmental and Chemical Engineering, Jiangsu University of Science and Technology, Zhenjiang 212003, China

**Keywords:** Na^+^ batteries, Li^+^ batteries, transition metals, dichalcogenides, structure engineering

## Abstract

Currently, transition metal dichalcogenides-based alkaline metal ion batteries have been extensively investigated for renewable energy applications to overcome the energy crisis and environmental pollution. The layered morphologys with a large surface area favors high electrochemical properties. Thermal stability, mechanical structural stability, and high conductivity are the primary features of layered transition metal dichalcogenides (L-TMDs). L-TMDs are used as battery materials and as supporters for other active materials. However, these materials still face aggregation, which reduces their applicability in batteries. In this review, a comprehensive study has been undertaken on recent advancements in L-TMDs-based materials, including 0D, 1D, 2D, 3D, and other carbon materials. Types of structural engineering, such as interlayer spacing, surface defects, phase control, heteroatom doping, and alloying, have been summarized. The synthetic strategy of structural engineering and its effects have been deeply discussed. Lithium- and sodium-ion battery applications have been summarized in this study. This is the first review article to summarize different morphology-based TMDs with their intrinsic properties for alkali metal ion batteries (AMIBs), so it is believed that this review article will improve overall knowledge of TMDs for AMIBS applications.

## 1. Introduction

The current rate of depletion of available energy sources and increasing demands for daily energy needs can easily lead to the conclusion that, in the near future, the world will require power sources of about 30 TW (10^12^) [1]. These energy crises and related environmental issues have created great pressure on researchers to develop sustainable and renewable sources for the fulfillment of modern society’s needs [2,3,4,5,6]. In this regard, fuel cells, solar cells, biomass, and wind-based energy sources are playing significant roles. However, a low-cost, environmentally benign, and highly efficient system is required for the storage and complete utilization of these renewable sources of energy [7,8]. The most appropriate energy storage system (ESS) is a battery because of its low maintenance, long cycle life, and high efficiency for energy conversion [9,10,11,12]. In this regard, during the last two decades rechargeable Li-ion batteries have been extensively studied for daily life uses such as cell phones, hybrid electric vehicles, and laptops [7,9,13,14,15,16]. However, materials in greater amounts are needed to develop secondary batteries for the construction of this smart-grid ESS. As a result, these primary aspects, including cost, power density, energy density, life, and safety, must be well balanced for these secondary batteries [17,18,19,20]. Less quantity (20 ppm) of Li in the earth’s crust, continuous demand for LIB, and raising prices hinder its practical application for energy storage systems (ESS) [21,22,23,24,25]. However, the similar chemical behavior of K, Na, and Li, along with their abundance, low price, and small atoms, give them a competitive edge over lithium for their use as active materials in potassium-ion batteries (PIBs) and sodium-ion batteries (SIBs) as ESS [26,27,28,29,30,31]. The rocking-chair mechanism of these LIBs, SIBs, and PIBs causes a reversible exchange of alkali metals between electrodes, where the anode plays a primary role in the performance of batteries. Thus, abundant, cheap, stable, and eco-friendly anode materials are of primary interest for cost-effective, extensive, and long-life applications [32].

The high conductivity, flexibility, surface area, and functionality with other molecules provide graphene special potential to be used in a supercapacitor, fuel cells, photovoltaic devices, and LIBs [33,34,35,36]. The layered morphology of graphene with such great potential gives motivation to researchers to study layered morphology-based inorganic materials, such as (L-TMDs) SnSe_2_, SnS_2_, MoSe_2_, MoS_2_, and WS_2_ [37,38,39,40,41]. L-TMDs with the MX_2_ formula (M = Hf, Ti, Zr, Mo, Sn, W, V Ta, Nb, etc., and X = Te, Se, S, etc.) exhibited graphite-like morphology and performances. The X atoms are arranged in the outer layer, which sandwiches the hexagonally closed-packed arranged metals as X-M-X, where the M-X bond is covalent and the bond between layers is van der Waals force. Their crystalline structure varies from orthorhombic (WTe_2_) to hexagonal (WS_2_). This arrangement and morphology give prime attention to L-TMDs to be applied in the field of energy conversion and storage devices due to their distinctive electronic, chemical, and physical properties [42,43,44,45,46,47,48,49,50,51]. The intercalation of alkali metal ions (K^+^, Na^+^, Li^+^) is supported by interlayered weak van der Waals forces [46,47,52]. Whittingham’s seminal work showed that Li ions can be reversibly inserted/extracted in the layers of TiS_2_ at a high potential of 1.5 V with a high speed of diffusion [53,54]. Similarly, 2D-base MoS_2_ showed high efficiency as a host for the insertion of Mg ions [29]. In addition, the conversion of one atom of MX_2_ requires four atoms of alkali metal, which exhibits a high theoretical capacity and has therefore been extensively studied as auspicious anode materials for AIMBs [55,56,57,58]. However, these anode materials still face challenges such as aggregation and volume change during charge/discharge, which result in low cycling life [32,59,60]. The electrochemical performance of anode material may be enhanced through phase control, surface defects, interlayer spacing expansion, and especially alloying and heteroatom doping to overcome the above problems [60,61,62,63,64,65,66].

In this review, the synthesis, modification, and application of sodium- and lithium-ion batteries have been summarized. In the first part, synthetic approaches for the development of 0D, 1D, 2D, and 3D morphology-based TMDs as electrocatalysts for AIMBs are discussed. In the second part, intrinsic modification and/or structural engineering, which can further improve the battery application of TMDs, have been discussed. In the last part, the electrochemical applications of lithium and sodium ion batteries are discussed.

## 2. Preparative Strategies for L-TMDs Based Composites

The active and exposed surface of anode materials governs the charge storage competencies for alkali metal ion batteries. The 2D morphology and large surface area of the L-TMDs make them special materials for ESS due to their large exposed surface area. The van der Waals force causes restacking of nearby 2D layers during battery cycling processes, which reduces the surface area of the 2D morphology and diminishes its significance with respect to electrochemical applications. In this regard, the most efficient approach is the hybridization of other proficient materials with L-TMDs to retain their unique morphology and characteristics on a larger surface area with an enriched electrochemical response. Recently, the hybridization of TMDs has been extensively studied and developed with 0D nanoparticles (NPs), 1D, 2D, and 3D morphologies.

### 2.1. 0D (NPs) Based L-TMDs Composites

During reversible storage of charge, L-TMDs show restacking of layers to reduce active surface area and conductivity, which limit their electrochemical applications. The incorporation of 0D NPs, such as metal/metal oxide or noble metal, in between the layer of TMDs will offer charge carriers that may boost their conductivity and proficiently hinder the restacking of layers. This is one of the possible ways to control the surface area and improve the conductivity of L-TMDs during the cycling process [67,68,69,70,71]. The incorporated NPs are used as spacers to support the materials and to allow the access of electrolytes for alkaliation/de-alkaliation between the active layers of TMDs. This improves their electrochemical application for SIBs and LIBs. Recent studies showed that the amalgamation of noble metals (Pt, Au, Pd, and Ag) has been determined to be the best option for enhancing the intrinsic conductivity of L-TMDs [72,73,74,75]. In this regard, Li et al. [67] obtained MoS_2_-C nanosheets from thiourea, Na_2_MoO_4_·2H_2_O, and glucose as raw materials through hydrothermal and Ar atmospheric pyrolysis. The as-obtained materials were again subjected to hydrothermal reactions with Sn precursors, followed by Ar calcination to obtain a MoS_2_ layer embedded with Sn NPs. These NPs work as spacers to avoid agglomeration and improve the electrochemical performance of the materials. Similarly, Chen et al. [70] decorated nanosheets of MoS_2_ with ultrasmall NPs of Fe_3_O_4_ via a two-step hydrothermal process. Their study concluded that the flexible and elastic features of MoS_2_ NSs make it a suitable substrate, as compared to reduced graphene oxide (rGO), for electrochemical applications. As well known, L-TMDs tend to agglomerate and become crumpled. Thus, the use of Fe_3_O_4_ NPs will act as spacers for the sheets of MoS_2_, hinder their collapsing during the charge/discharge processes of batteries, and enhance their cycling performance and reversible capacity. While Pan et al. [72] used a multifunctional organic ligand to anchor Ag NPs onto nanosheets of MoS_2_ via coordination. The NPs of Au were used as spacers to stop the restacking of layers of MoS_2_, and their high characteristics of conductivity also boosted the conductivity of the study materials. In addition, the composite structure remained effectively conserved due to the low deformability and high inflexibility of Au NPs. However, it is well known that the performance of bimetallic materials is better than monometals. In this regard, the reversible capacity and high conductivity of monometallic materials may be further enhanced via the co-doping of bimetals in L-TMDs. For instance, Pan et al. [71] used a one-step scheme to prepare a ternary heterostructure (Ag/Fe_3_O_4_/MoS_2_). In this research study, the author used THF (tetrahydrofuran) as the reaction medium and mixed MoS_2_ NSs with Ag and Fe_3_O_4_ NPs under magnetic stirring. The surfaces of the MoS_2_ layers were spontaneously decorated with metal NPs due to their van der Walls bond (Figure 1). The intrinsic conductivity and lithium storage reversibility were enhanced correspondingly by Ag and iron oxide NPs. Although there have been extensive studies, controlling the number of layers, especially monolayer TMDs, during hydrothermal synthesis is still a big challenge and a hot topic for research among scientists. This is because it not only offers an additional and transformational site for Na and Li storage but also frequent interfacial Na and Li storage sites [60].

### 2.2. 1D Based L-TMDs Composites

The L-TMDs NSs were also hybridized with 1D morphology in the anode materials in order to improve their reversible charge storage. In this regard, multiwalled carbon nanotubes (MWCNTs) and/or single-walled carbon nanotubes (SWCNTs) attract extensive research attention to be doped with L-TMDs in the field of electrochemical application based on their excellent electronic and mechanical properties. The CNTs are considered an excellent substrate to be used for L-TMDs NSs growth because, with their 1D nature, they offer a good pathway for the conduction of electricity during the cycling process [76,77,78,79]. Furthermore, during the charge/discharge process, the materials of the electrode cause expansion that can be accommodated by CNTs due to their great flexibility. In this regard, extensive research has been made to develop L-TMDs NSs decorated on CNT as 1D electrode materials. Ding et al. [79] used Na_2_MoO_4_·6H_2_O, glucose, and thiourea to develop CNTs ornamented with MoS_2_ NSs via a simple hydrothermal method. The synthesis method confirmed that the surface of CNT is homogeneously covered with MoS_2_, and glucose pyrolysis produces a well-developed conductive network. The data also confirmed the excellent interaction of MoS_2_ NSs with the CNT backbone. Recently, biomolecular synthetic strategies have been used to prepare different effective nanostructure-based materials [78,80]. The special structure of L-cysteine is due to numerous functional groups, such as -COOH, -SH, and -NH_2_, which make it a promising material among different biomolecules. L-cysteine can be used as a source of S during the hydrothermal synthesis process. In this regard, CNTs were treated with acid and then hydrothermally reacted with L-cysteine and sodium molybdate dehydrate and finally annealed to obtain CNTs-MoS_2_ as the required materials [78]. The ornamentation of SnS_2_ on MWCNT is due to the electrostatic interaction of the Sn^4+^ ion with the negative charge of MWCNTs, which is followed by the nucleation of SnS_2_ for the growth of 2D morphology as shown in Figure 2 [77]. In addition to TMDs-based hybrid composites, multi- and/or single-layered CNT, carbon nanofiber (CNF), nitrogen-doped CNTs (N-CNTs), and N-CNF embedded with MoS_2_ have also been extensively studied and reported [81,82,83]. Kong et al. [81] developed MoS_2_ embedded in graphitic CNTs; the hollow CNTs are made up of rolled graphene that hold MoS_2_ inside the CNTs. One of the excellent strategies for developing single-layer NSs of L-TMDs is the electrospinning method, in which nanoplates of WS_2_ with a single-layer morphology have been developed via electrospinning and then homogeneously ornamented on N-CNF [83].

Similarly, 1D structures of non-carbon substrate have also been used to decorate the NSs of L-TMDs. Nanotubes/nanobelts/nanowires of TiO_2_ have been used as substrates for the decoration of MoS_2_ NSs via solvothermal and/or hydrothermal strategies [84,85]. Li et al. [86] developed NSs of MoS_2_ on the surface of TiO_2_ nanowires from glucose via hydrothermal, which is a cost-effective route. The NSs of MoS_2_ were grown and nucleated on TiO_2_, which act as an effective template due to its robust nature, resulting in a nanocomposite with hierarchical morphology. The homogeneity of MoS_2_ NSs in the obtained composite is attributed to the roughness features of glucose and TiO_2_. However, one of the biggest challenges is achieving controlled and uniform deposition of MoS_2_ on a TiO_2_ surface due to their chemical reactions and poor affinities.

### 2.3. 2D Based L-TMDs Composites

The accommodation of L-TMDs on the surface of a 2D substrate may further enhance their excellent performance in electrochemical applications. The honeycomb-like morphology and π-conjugation, along with its extraordinary electrochemical performance, make graphene a tremendous substrate material for the decoration of L-TMDs. Graphene also exhibits high flexibility, outstanding mechanical properties, high conductivity, and a large surface area. The hybridization of L-TMDs with graphene may further boost their electrochemical applications due to their well-developed inherent conductivity, which provides a better pathway for the conduction of electrons, increases the electrochemical active surface area, facilitates the diffusion of mass, and improves the electrolyte interface. Graphene-based multi- and/or single-layered L-MoX_2_ (X = S, Se) composites can be easily synthesized via wet chemical or surfactant-assisted hydrothermal methods [87,88,89,90]. Change et al. [80] developed a MoS_2_/graphene hybrid composite via a novel synthetic approach (the L-cysteine-abetted solution-phase method). The authors used GO with L-cysteine and sodium molybdate as starting materials, followed by an annealing treatment to obtain the final composite. Similarly, MoS_2_/GNS hybrid materials have been synthesized via a wet chemical approach. In this method, hydrazine was used to reduce GO sheets, followed by the addition of (NH_4_)_2_MoS_4_ in the presence of cetyltrimethylammonium bromide (CTAB). The resulting mixture was then heated in an N_2_ atmosphere to obtain the required product [90]. Other cation-based surfactants, such as dodecyltrimethylammonium bromide (DTAB), tetrabutylammonium bromide (TBAB), and octocyltrimethylammonium bromide (OTAB), have also been applied during the synthesis of layered MoS_2_/GNS composites [91]. The type and concentration of cationic surfactant not only control the resulting product morphology but also determine the number and modified nature of MoS_2_ layers, thereby controlling the electrochemical performance of hybrid composite materials [92].

The wrinkled and highly rough surface of nitrogen-doped graphene (NG) exhibits a great number of defected sites, including an increased number of vacancies, which enhanced electrochemical reactions. These features of NG make it a suitable substrate for the growth of nanomaterials, allowing for the development of electrodes with nanomorphology. Cheng et al. [93] used a two-step approach for the synthesis of the MoS_2_/NG composite. In the first step, the author used elevated temperature to react L-cysteine as a source of S with NG for doping of S, followed by the nucleation growth of NSs of MoS_2_ as shown in liquid phase exfoliation (alongside with hydrothermal/wet chemical strategy). This is a scalable approach that can be applied to the synthesis of MoS_2_-rGO composites. According to this technique, NSs of MoS_2_ were first synthesized in the presence of PVP via hydrothermal method at low temperature. In the next step, an ultrasonic exfoliation approach was applied to obtain MoS_2_-rGO composites with a flower-like morphology [94]. MoS_2_ with a maximum of 1~3 layers was embedded in rGO in layer-by-layer morphology through exfoliation and chemical lithiation strategies to develop MoS_2_/rGO hybrid composites [95].

The electrochemical application of SnX_2_ (X = S, Se) base L-TMDs has also been improved through their hybridization with graphene. The SnX_2_ has also been synthesized through the same approach (the wet chemical method) and as applied for the preparation of MoX_2_ [96,97,98,99]. The method of solution chemistry was used to prepare a few-layer SnS_2_/graphene (FL-SnS_2_/G)-based composite in the presence of L-cysteine, which acts as a source of sulfur and as a reducing and complexing agent [100]. SnS_2_/rGO with a morphology of sheet-on-sheet has also been synthesized via a 1-step microwave-supported technique that exhibited highly improved photocatalytic and electrochemical applications, as shown in Figure 3 [101]. In addition, the SnS_2_-graphene-based composites have also been developed through the CVD method. The synthetic approach for 2D graphene-based SnS_2_ composites consisted of two steps: in the first step, graphene layers were decorated with NPs of SnO_2_ through a solution method, which was then converted to SnS_2_ via a simple CVD technique [102]. Similarly, WS_2_ supported on the surface of graphene layered (WS_2_-graphene composite) has been developed and deeply studied for electrochemical applications. WS_2_ base composite exhibited higher conductivity with better electrochemical activity as compared to MoS_2_ [103,104,105]. A solution-phase approach for the development of a few layered FL-WS_2_-rGO has been reported, in which the hydrazine hydrate is used to reduce GO into rGO under reflux [106].

### 2.4. 3D (2D Graphene) Based L-TMDs Composites

Although graphene-based L-TMDs nanocomposite demonstrated better electrochemical application, however, during the battery’s cycling process, the NS layers of L-TMDs and graphene cause aggregation due to their non-intimate interaction between L-TMDs and graphene, which later causees contact resistance among various reactive NSs. Recently, CNF and graphene-based MoX_2_ (X = Se, S) with 3D morphology have been reported [107,108,109,110]. NSs of GO and MoS_2_ are used as building blocks for the synthesis of 3D porous morphology by hydrothermal reaction based on a bottom-up synthetic route, and the composite demonstrated that NSs of MoS_2_ were restricted in 3D graphene as shown in Figure 4 [111]. In addition, oxidized MWCNT (o-MWCNT) along with graphene were simultaneously applied with MoS_2_ for the synthesis of 3D flower-like MoS_2_ (f-MoS_2_) for the hierarchical morphology of the f-MoS_2_/RGO/o-MWCNT composite [109]. The flower-like morphology of the composite may be attributed to the addition of o-MWCNT, which also avoided restacking of rGO. In addition to MoS_2_, the hydrothermal synthetic approach was used for the first time to develop MoSe_2_/rGO foam with 3D mesoporous morphology and apply it to LIBs as anode materials [110]. Similarly, the electrochemical application of composites consisting of WS_2_ and SnS_2_ with porous 3D morphology has also been studied, which showed improvement in their performance after their hybridization [112,113,114]. Recently, L-cysteine, SnCl_4_·5H_2_O, and NSs of GO were used as starting materials to develop SnS_2_ nanosheets homogeneously in the 3D framework of graphene. The author used two steps for the synthesis of composites: in the first step, the water bath method was used to decorate NPs of SnO_2_ on 3D graphene [115]. In the second step, L-cysteine was used as sulfur and a reducing source to convert SnO_2_ into SnS_2_ [116].

Similarly, Huang et al. used the low temperature hydrothermal method to decorate WS_2_ on graphene with interconnected 3D morphology; the author applied the composite materials for hydrogen evolution reactions and LIB [117]. In addition to the carbon-based 3D structure of MoS_2_, recently TiO_2_-supported MoS_2_ with 3D morphology has also been synthesized, and their data showed improved electrochemical response for LIBs. A hydrothermal method consisting of two steps was used to embed the NSs of MoS_2_ in firework-shaped TiO_2_ (F-TiO_2_) microspheres, which act as templates. The composite material was used as an anode material for LIBs and showed improved electrochemical performance [118].

### 2.5. 3D (Bulk Materials) Based L-TMDs Composites

An anode material may also be developed using 3D graphene instead of 2D graphene. The best method for the successful synthesis of 3D graphene networks (3DGNs) is the CVD technique; the as-synthesized 3DGNs can act as substrates for the growth of functional nanomaterials to develop active and efficient composites [119,120]. The 3DGNs exhibit both features, such as excellent conductivity and a high surface area of graphene. A precursor solution of (NH_4_)_2_MoS_4_ in dimethylformamide (DMF) was used to coat flakes of MoS_2_ on the surface of 3DGNs through a CVD synthetic approach [121]. Similarly, activated carbon fiber (ACF) was used as supporting material to deposit NSs of MoS_2_. Wang et al. used pre-annealed ACF and DMF solutions of (NH_4_)_2_MoS_4_ as precursor materials to synthesize ultrathin NSs of MoS_2_ on an ACF surface through a two-step mechanism. The precursors of ACF and DMF solutions were subsequently annealed in an atmosphere of 5% H_2_/Ar [122].

### 2.6. Non-Crytalline Carbon Based L-TMDs Composites

The L-TMDs have also been decorated on the surfaces of amorphous carbon [123,124,125]. Chang et al. [126] used a simple hydrothermal method and then applied calcination at high temperatures to develop MoS_2_/amorphous carbon-based composite. Similarly, glucose, Se powder, and Na_2_MoO_4_·2H_2_O were used as precursors to synthesize MoSe_2_/C composite with sheet-like morphology through a solution-phase strategy followed by heat treatment. The carbon not only hinders the stacking of sheets but also maintains the integrity of the structure [127]. Recently, NSs of L-TMDs have also been grown on the surfaces of hollow carbon structures such as porous hollow carbon spheres (PHCS) and N-doped hollow carbon nanoboxes (N-HCNB). These N-HCNB and PHCS not only reduced the aggregation of NSs in L-TMDs but also boosted nanocomposite conductivity during battery application [128,129]. A four-step method was used to decorate MoS_2_ on N-HCNB. The N-HCND can accommodate the strain, hinder the aggregation of NSs of MoS_2_, and enhance the composite conductivity during battery application and cycling processes.

## 3. Structural Engineering in L-TMDs

### 3.1. Interlayer Spacing Engineering

Van der Waals (vdW) force/interaction play a vital role in diffusion energy, as ReS_2_ has a lower vdW as compared to MoS_2_. Thus, lithium ion diffusion energy is lower for ReS_2_ as compared to MoS_2_, even with the same interlayer spacing. This vdW can be reduced by expanding the interlayer spacing, which also provides more sites for alkali ion storage with lower diffusion kinetics [130,131]. It is confirmed that increasing the interlayer space from 13 to 18 Å can reduce the diffusion barrier from 1.14 to 0.20 eV for sodium ions in MoS_2_ [132]. The same response is obtained for lithium and potassium ions [133]. Bottom-up or top-down-based strategies such as solvothermal, exfoliation-restacking, chemical vapor transport, and template intercalation can be used for interlayer spacing. In the solvothermal process, foreign species such as ammonium ions, carbon, and oxygen are formed from the precursor, which plays a vital role in the interlayer space, because these species settle down on the surface of L-TMDs and create/increase interlayer space. However, it is important to know that the interlayers space is dependent on reaction temperature, time, and precursors, which can release these foreign species during the reaction [32]. In this regard, a study showed that, increasing the concentration of thiourea from 14 to 60 mmol, the interlayer space increased from 0.63 to 0.91 nm [134].

The same response has been recorded for carbon; as organic compound concentration decreases or reaction temperature (for oxygen also) increases, it may lead to a decrease in the interlayer space of L-TMDs [91,135,136]. Similarly, in bottom-up methods such as the chemical vapor transport method, reaction time, temperature, and precursor ratio play vital roles in the final interlayer space. The main reason for interlayer spacing in this method is carbon, which has the same response as in the solvothermal [132]. In the exfoliation method, the layers of L-TMDs are exfoliated, and organic polymer is introduced in between. The polymer adsorbs on the surface and acts as an intercalation agent to increase interlayer space [137]. Ammonium ions or octylamine can be used instead of polymers to increase interlayer space. Octylamine can absorb polyaniline, while ammonium ions can be replaced with other organic compounds, such as sugar, which produce carbon during the annealing process [138,139]. A general scheme for interlayer spacing engineering is shown in Figure 5.

### 3.2. Surface Defect Engineering

A previous study confirmed that long diffusion pathways, insufficient channels, and high diffusion energy strongly reduce the battery application of L-TMDs. Surface defect engineering is the best approach to reduce these issues and improve L-TMD’s application for batteries. Surface engineering improves diffusion kinetics as it provides an extra pathway for alkali ion diffusion. TMDs-based catalysts (MoS_2_) exhibited seven kinds of surface defects, such as S single vacancy, Mo single vacancy, S double vacancies, a vacancy complex of Mo and three nearby S_2_ pairs, antisite defects where Mo substitutes S, a vacancy complex of Mo and nearby three S, and S substituting Mo [140]. Surface defects are developed in two ways. Post-treatment such as hydrogen/oxygen treatment or modulating solvothermal methods can be used to create a surface defect to facilitate an insertion channel, which ultimately reduces the alkali ion diffusion pathway and improves its electrochemical application [141,142,143]. In this regard, MoS_2_ was heated in air at a high temperature to oxidize its surface and create vacancies with grain boundaries. Surface-defect MoS_2_ showed better kinetics with high current density for sodium ion diffusion as compared to un-defect MoS_2_ as shown in Figure 6. Surface defects also improve capacitance and durability [144,145]. On the other hand, adsorption factor, reaction time, and temperature played vital roles in the surface defect of active materials during the solvothermal strategy. Thiourea is the most commonly used material for adsorption factors [32,134,146,147,148]. The main concept behind these defects is the adsorption of thiourea on the surface of active materials, which hinders further attachment of active material to the nucleation center. It may attach to the active spot of the materials during synthesis, and later, during the annealing process, it may dissociate, release, and leave a defect in the structure [149].

MoS_2_ with defect-rich synthesis via the solvothermal method showed a large surface area for lithium ion diffusion with excellent recyclability and rate performance. The research study also confirmed that 1,6-hexanediamine and carbon disulfide have also exhibited an adsorption-controlling effect for surface defects [150,151]. It is worth noting that while this approach to surface engineering can increase defects, it can also adversely affect the electrical properties of catalysts. Therefore, a major challenge is to create a surface defect that balances ionic diffusion with high electrical conductivity.

### 3.3. Phase Controlling Engineering

Intrinsic properties also play a significant role in the electrochemical application of catalysts. Thus, phase control or conversion of TMDs from semiconductor to metallic not only increased their battery application but also improved their structural stability [152,153]. Phase control is also possible via solvothermal, chemical vapor transportation, exfoliation, and 2H phase template-based strategies. Generally, in solvothermal methods, 2H (semiconditive) phase energy is favorable as compared to 1T (metallic) phase [154].

In order to obtain the 2H phase, it is necessary to hinder phase transformation from 1T -> 2H or lower the ground energy state of 1T [155]. In this regard, ammonium ions, carbon, and graphene hinder the 2H phase and lead to the 1T phase, as confirmed by Zhou et al. Additionally, reaction temperature also plays a vital role as it increases the favorability of phase transformation from 1T to 2H [156,157,158]. While Ding et al. [159] confirmed that strong magnetic fields favor 1T phase as magnetic fields reduce 1T phase energy due to its paramagnetic characteristic. This study confirmed that the 1T phase increased from 24.7% to 100% as the magnetic field increased from 0 to 9 T, as shown in Figure 7. On the other hand, high reaction temperatures during chemical vapor transport reactions favor the 2H phase [160]. In the case of exfoliation and the 2H phase template method, the introduction of foreign species favors S vacancies, which lead to the 1T phase of TMDs [161].

### 3.4. Doping Engineering

The introduction of heteroatom to TMDs as doped atoms has also gained extensive research attention. Various atoms can be doped, which increases their conductivity and diffusion kinetics for ions.

Sun et al. [162] found that the adsorption of alkali metal significantly reduced from −1.24 V for doped base TMDs. Qin et al. [163] developed N-doped MoS_2_ from thiourea and MoCl_5_ via the annealing method. It is also reported that TMDs can be prepared first and then treated with ammonia at high temperatures to develop N-doped TMDs [164]. The solvothermal method can be applied for doping of S, Se, and other metals (Fe, Sr, Co, Ni, Cr, Mn, etc.) in TMDs [165,166,167,168,169]. Doping TMDs with heteroatoms improved their conductivity due to synergism and hybridization of the heteroatom’s orbital with TMD’s orbital. This hybridization and/or synergism increase TMD’s electronic cloud with a configuration that gives a metallic feature to the catalyst, as shown in Figure 8.

### 3.5. Alloy Engineering

The introduction of metal in TMDs under certain conditions increases its conductivity from semiconductor to metallic, and this improvement is due to the hybridization of TMDs’ orbitals with introductory metals’ orbitals which strongly modify their electronic configuration [32]. In this regard, Cai et al. [171] introduce Se into MoS_2_ to produce alloys via the solvothermal method. Similarly, Wang et al. [172] and Jia et al. [173] introduce Se to MoS_2_ via chemical vapor transport and an annealing method. The alloy-TMDs exhibited metallic properties, which increase their electrochemical battery applications. Alloying engineering in catalysts for battery application can also reduce ions diffusion energy for the pathway, as shown in Figure 9. This engineering may increase its functionality, as discussed before.

## 4. Prospective Application of L-TMDs Based Composite

Increasing demand for energy and depletion of resources cause energy crises, which cause a great challenge for researchers to store the available sustainable and green energy. Recently, facile and novel materials have been developed to meet the need for energy and resolve the challenges. In the field of energy storage devices, L-TMDs have led the way in materials research due to their excellent electrochemical performance and physical properties. Here, the potential application of L-TMDs’ based composites will be briefly reviewed for SIBs and LIBs performances.

### 4.1. Lithium-Ion Batteries (LIBs)

The source of power in portable electronic devices such as hybrid electric vehicles (HEVs), electric vehicles (EVs), notebooks, mobile phones, and laptops is rechargeable LIBs, which are extensively applied due to their durable cycle life, high energy density, and no effect on memory [2,3,10]. Graphite has also been extensively applied as anode materials for batteries, but its low rate capability and specific capacity (372 mAh g^−1^) hinder its practical application for the demand of HEVs and EVs. The performance of LIBs may be enhanced with the use of NSs of L-TMDs instead of graphite, as L-TMDs exhibit a high surface area and a specific capacity with a short distance for the reaction of Li^+^, which may play a key role in practical application. However, their low intrinsic conductivity, weak cycle life, deprived rate capability, and layers restacking during charge/discharge of the batteries are the serious challenges [175]. The best solution to these challenges and to improve rate performance, cycle ability, and reversible capacity of LIBs is the hybridization of 0D (metal, metal oxide, noble metal NPs), 1D (nanobelts, nanotubes, nanowire, CNT, CNF), 2D (TiO_2_, graphene NSs), and 3D (ACF, 3D graphene, 3D porous architectures) based materials with NSs of L-TMDs for their practical application.

In this regard, 2D graphene-based L-TMDs NSs composites have been extensively studied as anode materials for LIBs [102,103,104,105,106,176]. MoS_2_/G is a typical example of a 2D graphene-based L-TMDs NSs composite that is developed through a solution-phase synthetic route. The as-synthesized composite was used as anode materials for LIBs, and the data exhibited excellent specific capacity of 1100 mAh g^−1^ at a current density of 0.1 A g^−1^ with improved cycle stability up to 100 cycles. The data confirmed excellent rate capability with a specific capacity of 900 mAh g^−1^ at a current density of 1.0 A g^−1^ [80]. This is the synergism of GNS and L-TMDs, which have improved the electrochemical application of these composites. The role of GNS in the composites is to minimize the electrode contact resistance and maintain the electronic conductivity of the composites. In addition, the GNS absorbed the peak of induced stress during the cycling process due to their flexible nature.

In this regard, graphene-based SnSe_2_, SnS_2_, and WS_2_ composites have been applied as anode materials for LIBs with improved performance [97,98,99,100,101]. The conductivity of graphene may be further enhanced via doping strategies, which can increase the storage capacity of Li in the anode of batteries. In this regard, N-G is the high-conductive kind of GNS, which has also been applied to develop N-GNS-based MS_2_ (M = W, Mo) composites [93,104]. The doping of N causes defect sides in the graphene and maximizes the vacancies for Li^+^; the active spots are further increased during the cycling process, which may increase the cycling stability and capacity of the composite and improve their performance [93]. In this regard, the rate capability and reversible capacity for LIBs of MoS_2_/N-G are much higher than those of MoS_2_ and MoS_2_/G composite. This obviously confirmed that gradual increases in vacancies or defect-based active sites may facilitate a greater number of Li^+^ being inserted in NG based materials during batteries charge/discharge.

L-TMDs based on 2D graphene and/or porous CNF composites with 3D interconnected porous morphology have also been applied as anode materials for LIBs [107,108,109,110,111,112]. The porous morphology provides a high surface area and a cushion toward the volume expansion of NSs of L-TMDs during the process of charge/discharge. In addition, the porous morphology also facilitates the access of electrolytes and charge transfer at a rapid rate due to the increased and continuous channels. These features of the 3D interconnected porous morphology of materials will deliver outstanding cyclic performance, excellent specific capacity, and ultrahigh-rate capability. Recently, the hydrothermal method was applied to develop MoS_2_ on the surface of graphene with a 3D porous structure. The as-synthesized composite was used as an anode material for LIBs and showed high performance as it exhibited a retention capacity of 1200 mAh g^−1^ even after 3000 cycles. Additionally, the composite showed excellent rate capability at 140 °C with a capacity of 270 mAh g^−1^ [107]. Similarly, MoS_2_/3DGN was synthesized using the CVD method, and the MoS_2_ was deposited on the surface of 3DGN. The composite materials were applied in LIB as anode materials and delivered a specific capacity of 665 mAh g^−1^ at a current density of 0.5 A g^−1^. The sample was run for 50 cycles, and again their specific capacity was studied at 4 A g^−1^ of current density; the sample showed a specific capacitance of 466 mAh g^−1^, as shown in Figure 10 [121]. The hybridization of NSs of L-TMDs with CNT [77,78,79,177], CNF [81,82,83], and amorphous carbon [123,126,127,128,129] has been deeply studied for anode materials of LIBs. These carbon-based materials also exhibit high conductivity and a large surface area. The deposition of L-TMDs NSs on these supporting materials dramatically boosted their outstanding cycling stability, reversible capacity, and rate performance as compared to their individual counterparts. Kong et al. synthesized NSs of MoS_2_ embedded in a nanocable of graphene (MoS_2_@G), which provided excellent reversible capacity with outstanding cycling stability. After 160 cycles, the sample delivered a capacitance of 1150 mAh g^−1^ at a current density of 0.5 A g^−1^. The rate capability of the composite was outstanding, as it delivered 700 mAh g^−1^ at 10 A g^−1^. Amazingly, this hybrid composite exhibited an extended cycle life (700 cycles) with excellent capacity of 900 mAh g^−1^ at a current density of 5 A g^−1^ [81].

High expansion of volume and weak conductivity are the two prime issues of nanostructures that limit their practical application as anode materials for LIBs. These nanostructured anode based materials initially deliver a very high capacity for charge/discharge, but rapid loss of their capacity hinders their long-term utilization for practical application. During the process of lithiation/delithiation in the material of the electrode, SEI layer formation occurs, which expands the volume of materials and, as a result, the capacity of the anode is directly dropped from its initial capacity value. The synthesis of TMDs with a single layer or the deposition of TMDs on the surface of various carbon base supporters with a flexible nature may accommodate the strains, which may reduce the limitation of anode materials. The limitation of anode materials may also be minimized to facilitate the intercalation of ions during the cycling process. Single-layered TMDs exhibited a short way of diffusion for Li^+^, extra storage of interfacial Li, a large surface area, and can accommodate changes in volume. These features give superior advantages to single-layered TMDs over multi-layered TMDs [82,83]. Zhu et al. [82] used the electrospinning method to decorate a single-layered nanoplate of ultrasmall MoS_2_ on the CNF, which delivered outstanding performance for LIBs. The CNF-supported ultrasmall size and single-layered morphology of MoS_2_ were confirmed from the TEM images. At a current density of 0.1 A g^−1^, the composite exhibited charge/discharge capacities of 1267 mAh g^−1^ and 1712 mAh g^−1^, respectively. The composite at current densities of 10, 30, and 50 A g^−1^ deliver specific capacities of 637, 548, and 374 mAh g^−1^, correspondingly. After 1000 cycles, the composite delivered a capacity of 661 mAh g^−1^ (10 A g^−1^ after), which confirmed the outstanding stability of the sample. Obviously, several factors are involved in the enhancement of the electrochemical performance of CNF-based composites, such as providing a number of sites for interfacial storage and offering different sites to be used for conversion and insertion of Li storage. In addition, the CNF hinders the aggregation of NPs and restacking of TMD layers during the cycling process. Similarly, carbon-based supporter networks reduce volume expansion and facilitate Li^+^ access, and their high conductivity enhanced their batteries applications, which may be further enhanced with the help of N-doping in CNF.

In addition to carbon based supporting materials, metal oxide, metal, and noble metal have also been used as supporting materials to decorate MoS_2_ to develop anode materials for LIBs with improved performance [67,68,69,70,71]. The flexible and elastic nature and large surface area of the NSs of MoS_2_ make them an ideal supporter to hinder NPs restacking and cushion the change in volume due to the charge/discharge process. The presence of NPs in between the NSs of MoS_2_ acts as a spacer, which stops the restacking of NSs of MoS_2_ during battery cycling processes and may enhance the retention of capacity. In addition, the diffusion pathway of Li^+^ is short, which gives superior rate capabilities and capacities to anode materials [72]. Recently, MoS_2_-supported 1D TiO_2_ nanobelt/nanowire/nanotube-based composites have been extensively studied as alternative materials for LIBs anode [84,85,86]. The composite delivered improved stability for rate capacity and cycling process, which may be due to less change in volume during the process of cycling with a high operation potential of 1.7 V (comparative to Li^+^/Li). However, the loose surface contact that interconnected the nanostructure of MoS_2_ and 1D TiO_2_ may cause self-aggregation during the process of charge/discharge, and it may limit their performances and practical application. Moreover, a TiO_2_/MoS_2_-based hollow porous 3D structure has been synthesized to be applied as anode materials in LIBs. The composite MoS_2_ is homogenously supported on the surface of TiO_2_, which dramatically shortens the length of the diffusion path and improves the stability of the structure for Li^+^ and electrons. TiO_2_/MoS_2_ with 3D morphology demonstrated superior rate capability with excellent cycling performance [118,150]. Less than 4 layers of MoS_2_-based NSs were coated on the surface of NSs of TiO_2_ to develop a 3D core-shell nanostructure (3D FL-MoS_2_@TiO_2_), which was used as an anode material for LIBs [150]. The composite delivered a capacity of 713.7 mAh g^−1^ (0.1 A g^−1^) even after 100 cycles, confirming its stable cycling performance. Similarly, Chen et al. [178] synthesized MoS_2_ on the surface of a carbon nanotube and applied for LIBs. The MoS_2_/CNTs showed a high capacity of 800 mAh g^−1^ at a current density of 5 A g^−1^. The materials also showed excellent rate performance, and this excellent response was due to the CNT presence, which improved the overall conductivity due to the structural stability of the composite at the same time.

### 4.2. Sodium Ion Batteries (SIBs)

LIBs have the potential to solve the growing energy crisis and meet the energy needs of society by providing energy systems for portable electrical devices and EVs. Currently, LIBs fulfill the demand for electrical energy storage in emerging markets, but the Li availability in the Earth’s crust must be considered. The abundance of Li is low and, thus, high prices and non-renewable resources are the basic limitation of LIBs. In comparison to Li, the abundance of Na is high and its price is low. Additionally, Al can also be used as a current collector in SIBs, as there is no report on the binary alloy of Al and Na. These are the features that make the SIB a highly promising alternative battery for LIBs [25,26,179]. However, the theoretical volumetric and gravimetric capacity of SIBs is much lower due to their weak electrochemical response as compared to LIBs. These characteristics and limitations of Na may be due to their heavier ion as compared to Li^+^, causing sluggish diffusion in the bulk of the electrode. Similarly, the high expansion of volume may result in low stability for the cyclic process and rate capability. Secondly, the lower value of volumetric and gravimetric energy densities may be due to the lower operating potential caused by the lower ionization potential of sodium. The development of unique crystalline and porous structures that exhibited large insertion channels for the facilitation of sodiation/de-sodiation may help to overcome these limitations. Recently, terrific research has been collected to study the feasibility of SIB as an anode in the field of ESS. The research from recent years has recommended NSs L-TMDs-based composites as suitable materials for SIBs anodes. As with LIBs, graphene-based L-TMDs have also been applied as anodes for the study of SIBs with/without binder [87,94,96,180]. In this regard, Zhang et al. [87] developed MoSe_2_/graphene composites with hierarchical morphology and applied them as anodes in SIBs. The composite delivers specific capacities of 380 and 430 mAh g^−1^ at current densities of 1 and 0.5 A g^−1^, respectively. The synergism interaction between GNS and MoSe_2_ is attributed to this excellent electrochemical performance. Carbon-based materials ornamented with L-TMDs have also been studied for SIB applications [76,82,125,129].

Zhu et al. developed CNF decorated with nanoplates of single-layered MoS_2_ and applied them for SIBs electrochemical applications. The composite delivered excellent cyclic stability and rate capability during SIB’s anode, as at current densities of 0.1, 0.5, 1, 5, 10, 20, 30, and 50 A g^−1^, the composite delivered specific capacities of 854, 700, 623, 436, 331, 224, 155, and 75 mAh g^−1^, respectively. The composite exhibited good performance even after 100 cycles, and at current densities of 1 and 10 A g^−1^, the composite delivered specific capacities of 484 and 253 mAh g^−1^, respectively. This is the best composite for SIBs reported up to now, and the outstanding performance of composite can be dedicated to single-layered MoS_2_ with its ultrasmall morphology [82]. Similarly, a three step method was applied to the synthesized NSs of MoSe_2_-based PHCS composites for the study of electrochemical application of SIBs [129]. At a current density of 0.2 A g^−1^, the composite delivered a specific capacity of 580 mAh g^−1^. The specific capacity of the sample remains at 400 mAh g^−1^ at current density of 1.5 A g^−1^ even after 100 cycles. The stress was accommodated by NSs of MoSe_2_ on the PHCS composite during the process of charge/discharge, as shown in Figure 11. A conductive atmosphere in the composite was created by PHCS, while NSs of MoSe_2_ facilitate the path of Na in SIBs [129]. Similarly, carbon-based MoS_2_ composite [125] and MWCNT-based MoSe_2_ composite [76] have also been developed and applied as anode materials for SIBs and delivered excellent performance. MoSe_2_@MWCNT delivered a specific capacity of 459 mAh g^−1^ at a current density of 200 mA g^−1^ over 90 cycles. At the current rate of 2000 mAh g^−1^, MoSe_2_@MWCNTs showed a specific capacity of 385 mAh g^−1^. This performance confirmed that MWCNT-based L-TMDs are excellent composites for SIBs, and this is due to the special structure and chemistry of MWCNTs, which facilitate electron passage and avoid agglomeration to retain a high surface area.

## 5. Conclusions and Future Perspective

AMIBs are a promising choice to meet energy demand, overcome the energy crisis, and address environmental issues as they provide energy to portable devices. Recently, L-TMDs-based materials have been extensively investigated for AMIBs due to their high electrochemical performance, high conductivity, mechanical, thermal, and chemical stability as compared to their bulk materials. TMD-based 0D, 1D, 2D, and 3D composites showed excellent performance for AMIBs. TMDs exhibited excellent performance for AMIBs in 2D morphology due to their morphology and large surface area, which were available for electrochemical reactions. However, the battery applications of 3D TMD composites showed outstanding performance due to the low agglomeration or stacking of active materials, which increased the surface area for electrochemical reactions. Additionally, precise control and improvements in the intrinsic feature can further improve its properties for application. Surface defects can increase active spots for reactions and reduce diffusion kinetics to enhance TMD’s properties for batteries. Interlayers space may help to avoid agglomeration, which leads to the retention of a large surface area and again reduces the diffusion pathway and kinetics. Phase control, alloying, and doping improve its intrinsic properties, electronic cloud, and electronic configuration, which ultimately increase batteries applications. However, it needs a lot of care and optimization studies to control this surface engineering. In the future, these materials for battery application must be prepared with 3D morphology to avoid agglomeration and utilize their maximum surface area. It is also very important to carry out these surface engineering techniques under precise conditions to optimize engineering for improved applications. It is also recommended to focus on other L-TMD-based electrocatalysts to be developed for extended battery applications. Multimetals and multichalcogenides are needed to investigate the high performance of alkali metal storage, long recycling, and high rate capability. The 3D morphology, which provides a short pathway for diffusion, fast kinetics for diffusion, large surface area for reaction, and alkali metal storage, are needed to investigate with the required surface engineering to further improve its application. Other supporting materials, especially boron nitride or boron-doped carbon nitride with 3D morphology, are needed to be developed and applied with TMDs for composite synthesis and battery applications.

## Figures and Tables

**Figure 1 materials-16-02559-f001:**
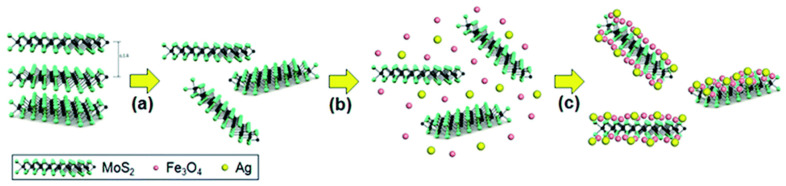
Schematic representation of (**a**) Liquid-phase exfoliation of MoS_2_ powder into nanosheets in NMP by sonication; (**b**) mixing of MoS_2_ nanosheets with Fe_3_O_4_@OA and Ag@ODA NPs (**c**) spontaneous hierarchical co-assembly of NPs on MoS_2_ nanosheets through van der Waals interactions. Reproduce with permission of ref. [71]. Copyright 2015, Royal Society of Chemistry.

**Figure 2 materials-16-02559-f002:**
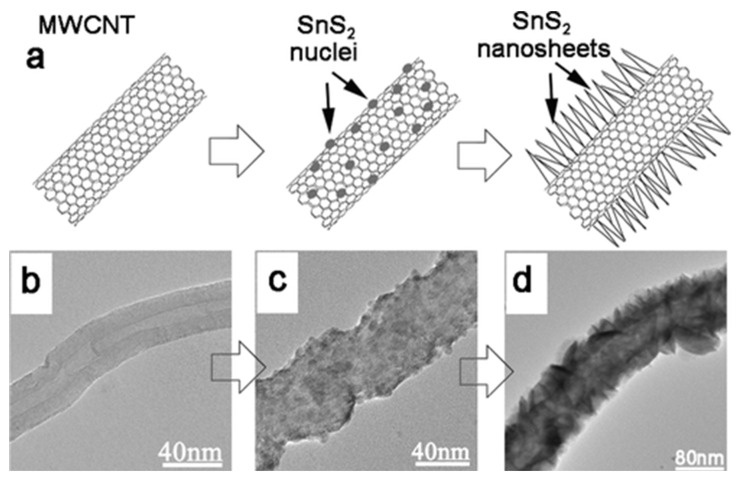
Schematic illustration and TEM images for the growth process of the SnS_2_ NS@MWCNTs coaxial nanocables: (**a**) schematic illustration; TEM image of the product sonicated at different times (**b**) 0 h (pure MWCNT); (**c**) 20 min; (**d**) 6 h. Reproduced with permission of ref. [77]. Copyright 2011. American Chemical Society.

**Figure 3 materials-16-02559-f003:**
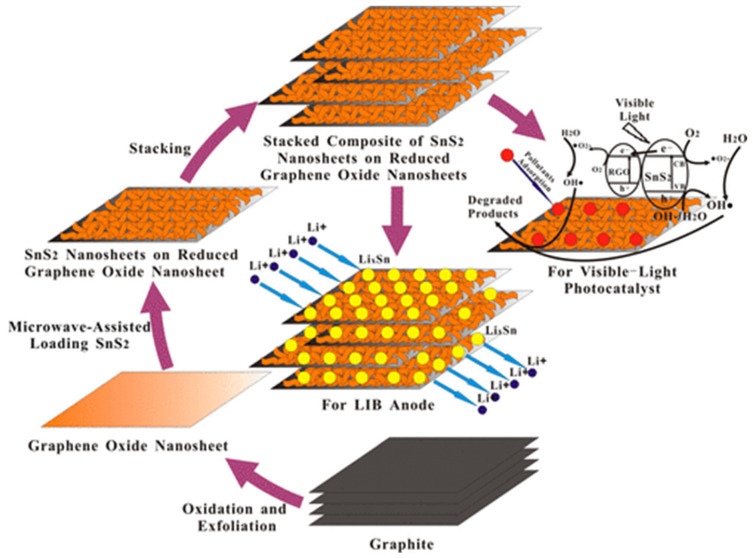
Schematic illustration of the growth process and applications of SnS_2_–RGO sheet-on-sheet nanostructure. Reproduced with permission of ref. [101]. Copyright 2013. American Chemical Society.

**Figure 4 materials-16-02559-f004:**
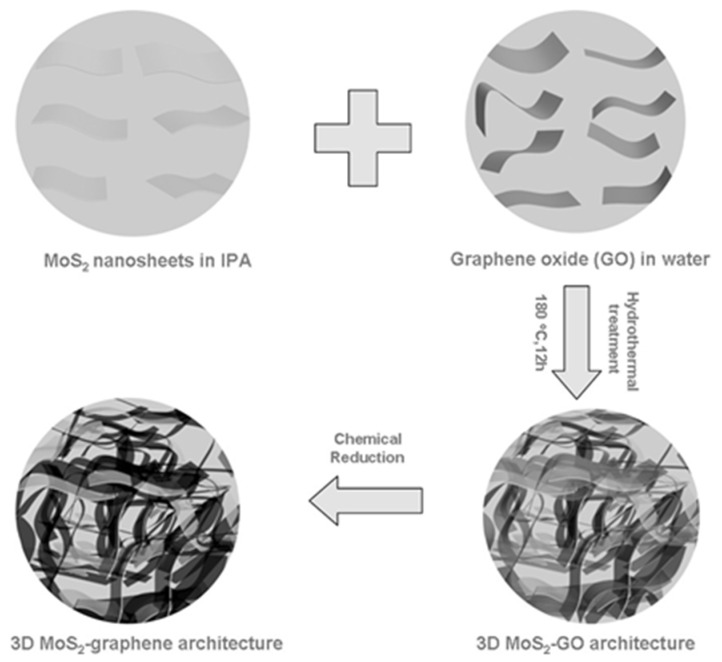
Schematic illustration for the construction of 3D MoS_2_-graphene architectures. It mainly involves three steps: (1) fabrication of MoS_2_ and graphene oxide nanosheets via liquid exfoliation and Hummers methods, respectively; (2) hydrothermal treatment of the mixed dispersion of MoS_2_ and GO nanosheets in IPA (isopropyl alcohol)/water (1:2, *v*/*v*) at 180 °C for 12 h; (3) chemical reduction of GO to generate 3D MoS_2_-graphene architectures. Reproduced with permission of ref. [111]. Copyright 2013 WILEY-VCH Verlag GmbH & Co. KGaA, Weinheim, Germany.

**Figure 5 materials-16-02559-f005:**
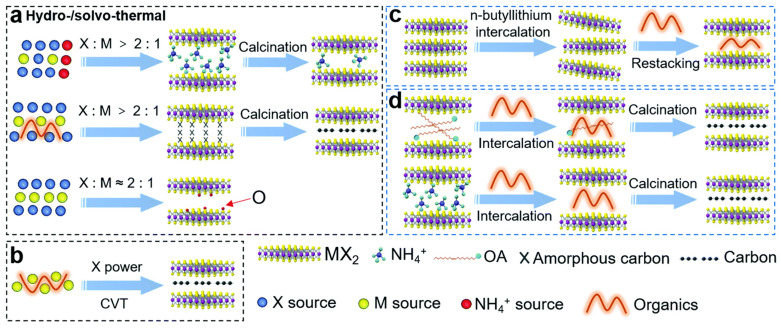
Schematic illustrations of methods for synthesis of the interlayer-expanded MX_2_. (**a**) hydro-/solvo-thermal; (**b**) chemical vapor transport (CVT); (**c**) exfoliation–restacking; (**d**) template intercalation method. Reproduced with permission of ref. [32]. Copyright 2020. Royal Society of Chemistry.

**Figure 6 materials-16-02559-f006:**
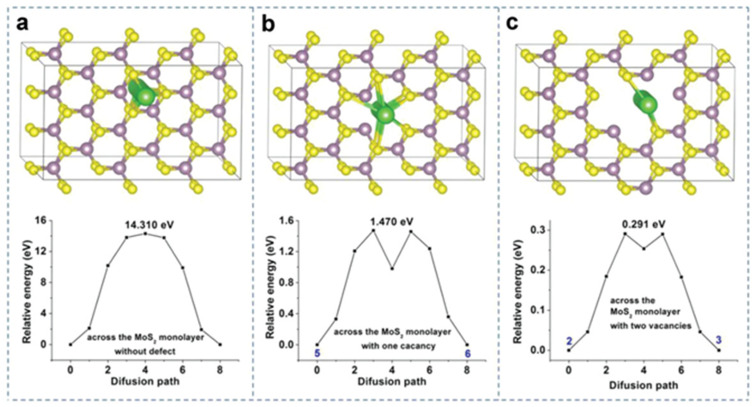
The pathways of the lowest Na^+^ diffusion barrier across (**a**) the indefective MoS_2_ monolayer, (**b**) defective MoS_2_ with one vacancy, and (**c**) two vacancies and their corresponding potential energy curves. The purple, yellow, and green balls represent Mo, S, and Na, respectively. Reproduced with permission of ref. [145]. Copyright 2019. WILEY-VCH Verlag GmbH & Co. KGaA, Weinheim.

**Figure 7 materials-16-02559-f007:**
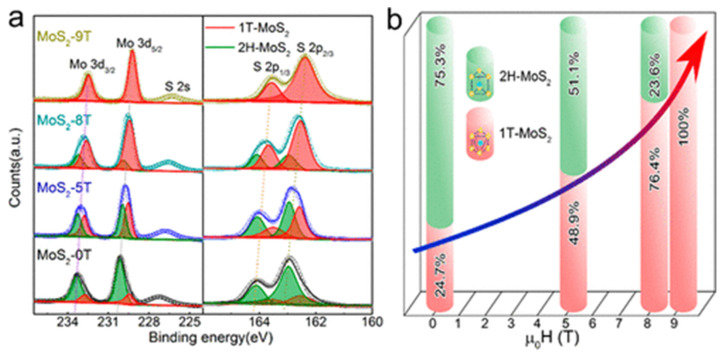
(**a**) XPS results of the as-synthesized MoS_2_ processed under different magnetic fields in magneto-hydrothermal processing. (**b**) Phase percentage of MoS_2_ from deconvolution of XPS spectra. Reproduced with permission of ref. [159]. Copyright 2019. American Chemical Society.

**Figure 8 materials-16-02559-f008:**
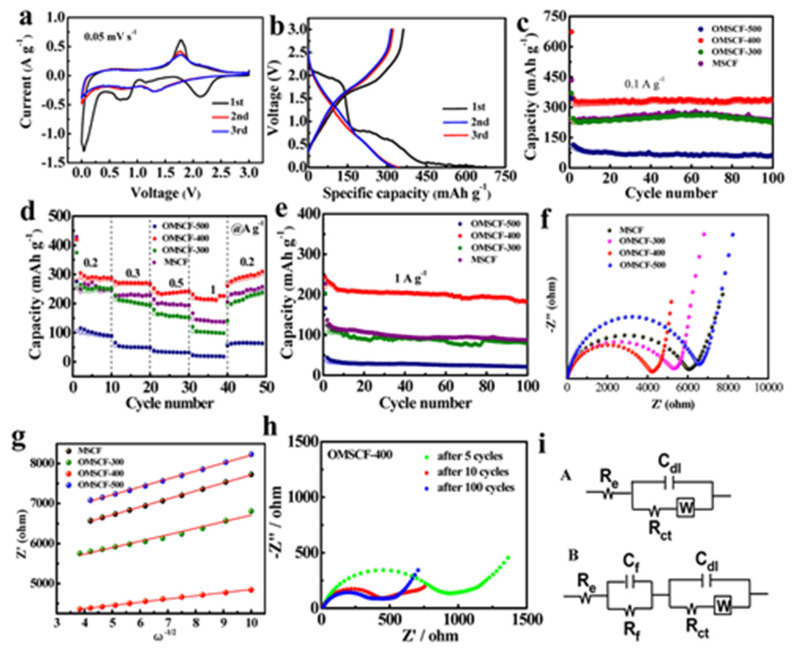
(**a**) CV curves of the OMSCF-400 electrode at 0.05 mV s^–1^. (**b**) Galvanostatic discharge–charge profiles of OMSCF-400 for the first three cycles at 0.1 A g^–1^. Cycling performance of MSCF, OMSCF-300, OMSCF-400, and OMSCF-500 at (**c**) 0.1 A g^–1^ and (**e**) 1 A g^–1^. (**d**) Rate performance of MSCF, OMSCF-300, OMSCF-400, and OMSCF-500. (**f**) EIS curves, and (**g**) the relationship between *Z*′ and ω^–1/2^ in the low-frequency range of MSCF, OMSCF-300, OMSCF-400, and OMSCF-500. (**h**) EIS of the OMSCF-400 electrode after various cycles. All impedance measurements were made at the fully charged state. (**i**) Equivalent circuit used to describe the Na^+^ insertion/extraction process for the electrodes (A) before and (B) after cycling. Reproduced with permission of ref. [170]. Copyright 2018. American Chemical Society.

**Figure 9 materials-16-02559-f009:**
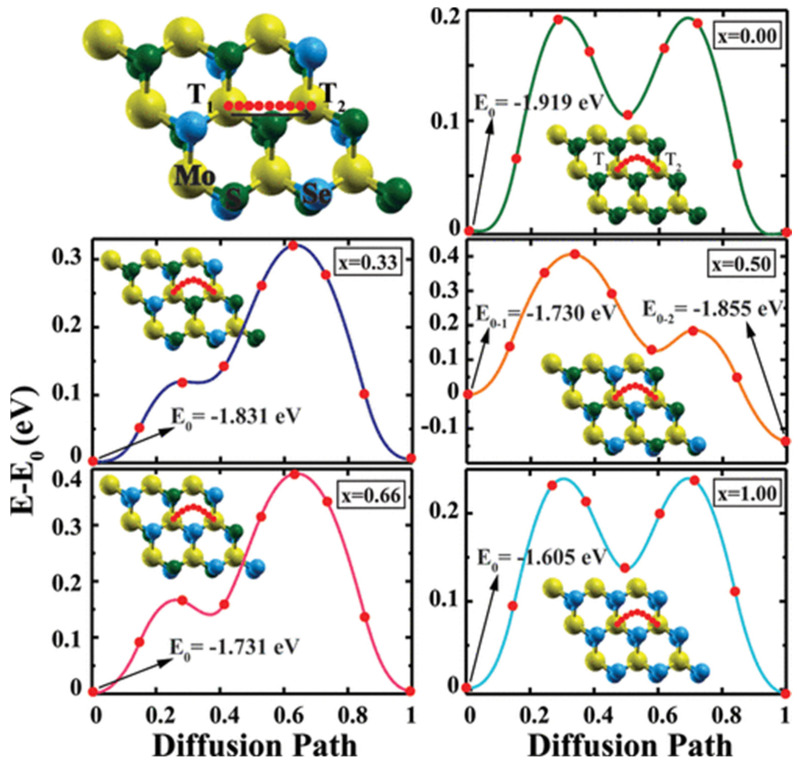
Lowest energy diffusion paths of a single lithium atom and calculated energy profiles along the paths for one Li atom adsorbed on the TMDs monolayer. Reproduced with permission of ref. [174]. Copyright 2015. American Chemical Society.

**Figure 10 materials-16-02559-f010:**
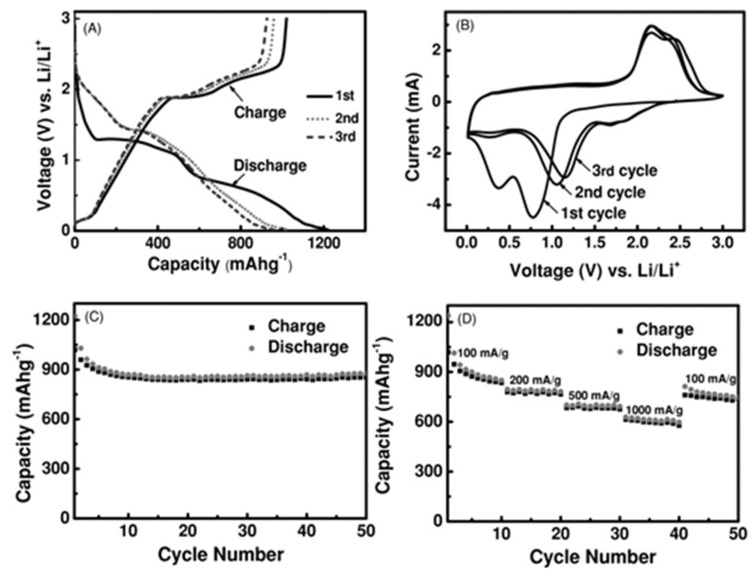
(**A**) Charge/discharge voltage profiles of MoS_2_/3DGN composite at the 1st, 2nd, and 3rd cycles. (**B**) Representative cyclic voltammograms of MoS_2_/3DGN composite at the first 3 cycles at a scan rate of 0.5 mV s^−1^ between 0.01 V and 3 V. (**C**) Cycling performance of MoS_2_/3DGN composites at a current density of 100 mA g^−1^. (**D**) Cycling stability of MoS_2_/3DGN composite at various current densities. Reproduced with permission of ref. [121]. Copyright 2013. WILEY-VCH Verlag GmbH & Co. KGaA, Weinheim.

**Figure 11 materials-16-02559-f011:**
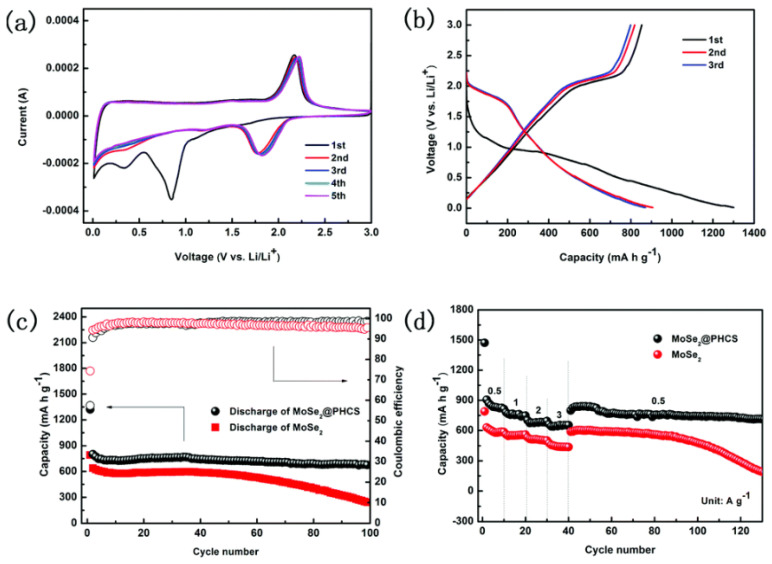
Electrochemical performance of the MoSe_2_@ PHCS composite and MoSe_2_ nanosheets for lithium batteries (the capacities were calculated based on MoSe_2_ mass). (**a**) Cyclic voltammogram profiles of the MoSe_2_@PHCS composite at a scan rate of 0.2 mV s^−1^. (**b**) Discharge/charge voltage profiles of the MoSe_2_@PHCS composite. (**c**) Cycling performance and Coulombic efficiency of the MoSe_2_@PHCS composite and MoSe_2_ nanosheets within a voltage range of 0.01–3.0 V. (**d**) The capacity of the MoSe_2_@PHCS composite and MoSe_2_ nanosheets under different current densities between 0.01 and 3.0 V. Reproduced with permission of ref. [129]. Copyright 2015. Royal Society of Chemistry.

## Data Availability

Not applicable.

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
