# Peer review of "Recent Advancement and Structural Engineering in Transition Metal Dichalcogenides for Alkali Metal Ions Batteries"

_materials, 2023, doi:10.3390/ma16072559_

Round 1

Reviewer 1 Report

This review summarizes the recent progresses in L-TMD based materials and is a good supplement to the field as it integrated different morphology base TMDs together. However, some issues exist hindering its suitability of publication at this stage. A minor revision is necessary. Below are some specific points that the authors need to take care.

1.      The English writing is not very good as obvious errors pop out frequently. The authors need to take a major care of their language.

2.      In the abstract, “AMIBs” is not defined at its first appearance.

3.      The subtitles are quite confusing. For example, why there is a point between 2D in “2.D base L-TMD composites” and what does “3. D (2D graphene) base L-TMD composites” mean as it contains both 3D and 2D?

4.      For battery applications, these L-TMD materials cannot be view as very practical due to their poor scalability of synthesis and very low volumetric energy density. It would be better if the authors can touch this point to give a perspective.

Author Response

Dear Editor/ Reviewer

We are highly thankful to the reviewer for showing significant interest in our research work. We appreciate the reviewer’s valuable contributions in providing best possible guidance and recommendations. The highlighted incorrect statements have been corrected as red colored text enclosed in quoted marks. All the mentioned points have been corrected with best alternatives and literature agreement. Our responses to reviewers’ comments are listed as follows.

Comment 1: The English writing is not very good as obvious errors pop out frequently. The authors need to take a major care of their language.

Response: Thank you for suggestion, the article is modified and English grammar is improved. 

Comment 2: In the abstract, “AMIBs” is not defined at its first appearance.

Response: Thank you for suggestion, The statement is modified in article,

Statement in manuscript:

with their intrinsic properties improvement for alkali metal ions batteries (AMIBs),

Comment 3: The subtitles are quite confusing. For example, why there is a point between 2D in “2.D base L-TMD composites” and what does “3. D (2D graphene) base L-TMD composites” mean as it contains both 3D and 2D?

Response: Thank you so much to mention this serious typo mistake, the article is modified according to suggestion.

Statement in manuscript:

0D (NPs) base L-TMDs composites

1D base L-TMDs composites

2D base L-TMDs composites

3D (2D graphene) base L-TMDs composites

3D (bult materials) base L-TMDs composites

Comment 4: For battery applications, these L-TMD materials cannot be view as very practical due to their poor scalability of synthesis and very low volumetric energy density. It would be better if the authors can touch this point to give a perspective.

Response: We agree with the reviewer's suggestion. However, the literature study has confirmed that 3D L-TMDs are the best option, and their performance can be further improved with surface engineering. Recently, the pyrolysis/annealing method and solvothermal methods have successfully overcome the synthesis scalability issues of these materials. Therefore, it is recommended to study 3D L-TMDs with optimized surface engineering for practical battery applications.

Statement in manuscript:

In the future, these materials for batteries application must be prepared with 3D morphology to avoid agglomeration and utilize its maximum surface area. It is also very important to carry out these surface engineering under precise condition to get optimize engineering with improve applications. It is also recommended to focus on other L-TMDs based electrocatalyst to be developed with better batteries applications. Multimetals and multichalcogenides are needed to investigate for high performance of alkali metal storage, long recycling, and high rate capability. 3D morphology which provide short pathway for diffusion, fast kinetics for diffusion, large surface area for reaction and alkali metal storage, are needed to investigate with required surface engineering to further improve its application. Other supporting materials especially, boron nitride, or boron doped carbon nitride with 3D morphology are needed to be develop and applied with TMDs for composite synthesis and for batteries applications.

Reviewer 2 Report

In this review, the authors have done a comprehensive study of layered transition metal dichalcogenides on Li and Na - ion batteries. The review is divided into three parts: synthetic approach, structural engineering, and electrochemical performance. In the synthetic part, authors have discussed the development of 0D, 1D, 2D and 3D morphology-based TMD electrocatalysts for AIMBs. In this section, the authors have explained the various synthesis approaches to increase the surface area and electric conductivity of 2D graphene and 1D MWCNT with the introduction of L-TMDs which affects the diffusion energies of these materials and decreases electrochemical properties. The second part demonstrates surface engineering via interlayer spacing, doping, alloy engineering, etc. which are responsible for affecting the properties of these materials and finally these synthesized materials are responsible for increasing battery performances. The review highlights the importance of these L-TMDs for these energy storage devices. However, there are some serious concerns that need to be corrected before considering this review for publication:

1.      The title of the article is completely unrelated to the comprehensive study of the review. There is not any description of alkaline metals. Instead, use alkali metals which demonstrate the study.

2.      The English of the review is very weak that needs to be corrected.

3.      Page 3, 6th line, ‘L-TMD [72-75]. in this regard’ – ‘In this Regard’

4.      Check the font size of the Figure 1 caption.

5.      Include the electrochemical studies of L-TMDS – MWCNTs electrodes in the electrochemical part.

6.      In the interlayer spacing engineering part, what is vdW? Explain it before mentioning its relevance. Also, in the line 5, ‘13 to 18 A’. What is A?

7.      Line 14, ‘In this Regard’

8.      In surface defect engineering, specify the part of thiourea more. Why is it important to?

9.   Phase controlling engineering, what is the combined effect of temperature and magnetic field?

10.  Doping engineering, what is the effect of this on diffusion energy?

11.  Check the subscripts.

Author Response

Dear Editor/ Reviewer

We are highly thankful to the reviewer for showing significant interest in our research work. We appreciate the reviewer’s valuable contributions in providing best possible guidance and recommendations. The highlighted incorrect statements have been corrected as red colored text enclosed in quoted marks. All the mentioned points have been corrected with best alternatives and literature agreement. Our responses to reviewers’ comments are listed as follows.

Comment 1: The title of the article is completely unrelated to the comprehensive study of the review. There is not any description of alkaline metals. Instead, use alkali metals which demonstrate the study.

Response: Thank you for your correction, it was a mistake and now the title changed to alkali metal.

Statement in manuscript:

Recent Advancement and Structural Engineering in Transition Metal Dichalcogenides for Alkali Metal Ions Batteries

Comment 2: The English of the review is very weak that needs to be corrected.

Response: Thank you for suggestion, the article is modified and English grammar is improved. 

Comment 3: Page 3, 6th line, ‘L-TMD [72-75]. in this regard’ – ‘In this Regard’

Response: Thank you for suggestion, the article is modified according to suggestion.

Statement in manuscript:

L-TMD [72-75]. In this regard, Li et al. [67]

Comment 4: Check the font size of the Figure 1 caption.

Response: Thank you for suggestion, the font size is modified according to suggestion.

Comment 5: Include the electrochemical studies of L-TMDS – MWCNTs electrodes in the electrochemical part.

Response: Thank you for suggestion, the required literature study is provided according to reviewer suggestion.

Statement in manuscript:

Lithium ions batteries,

Similarly, Chen et al. [178] synthesized MoS2 on the surface of carbon nanotube and applied for LIBs. The MoS2/CNTs showed a high capacity of 800 mAh g-1 at current density of 5 A g-1. The materials also showed excellent rate performance and this excellent response due to CNT which improved its conductivity due to their structure and also enhanced structural stability of the composite at same time.

Sodium Ions batteries,

MoSe2@MWCNT delivered specific capacity of 459 mAh g-1 at current density of 200 mA g-1 over 90 cycles. At current rate of 2000 mAh g-1 MoSe2@MWCNTs showed specific capacity of 385 mAh g-1. This performance confirmed that MWCNTs based L-TMDs are excellent composite for SIBs and this is due to special structure and chemistry of MWCNTs which facilitate electron conduction and avoid agglomeration to retain high surface area.

Comment 6: In the interlayer spacing engineering part, what is vdW? Explain it before mentioning its relevance. Also, in the line 5, ‘13 to 18 A’. What is A?

Response: Thank you for comment, vdW is standing for van der Wall force/interaction, and same information is provided in manuscripts as well.

Sorry for typo, it is Å not A, it is also correct in manuscript.

Statement in manuscript:

Van der Wall (vdW) force/interaction play vital role in diffusion energy,

It is confirmed increasing interlayer space from 13 to 18 Å can reduce diffusion barrier from 1.14 to 0.20 eV for sodium ion in MoS2 [132].

Comment 7: Line 14, ‘In this Regard’

Response: Thank you for correction, the manuscript is modified according to suggestion.

Statement in manuscript:

In this regard, a study showed that, increasing concentration of thiourea from 14 to 60 mmol, the interlayer space increased from 0.63 to 0.91 nm [134].

Comment 8: In surface defect engineering, specify the part of thiourea more. Why is it important to?

Response: Thank you for your suggestion, the manuscript is modified according reviewer to suggestion. Actually thiourea interacts with surface of active materials and settles down there which stop the attachment of incoming active particle. But during annealing process the thiourea is dissociate and release but the point where it was attached and it stop the incoming active materials is not bare and defected. For example we have a materials AB and A attached to B and B attached to A so if at one point we stop the attachment of A to B at that point B will be defected because A is absent, so same happen here.

Statement in manuscript:

The main concept behind these defects is the adsorption of thiourea on active materials surface, which hinders further attachment of active material to the nucleation center. It may attached to active spot of the materials during synthesis and later, during annealing process it is dissociate, release and left a defect in the structure [149].

Comment 9: Phase controlling engineering, what is the combined effect of temperature and magnetic field?

Response: it is very interesting question and matter of research, we are really sorry to say that up to our knowledge we didn’t find related research to understand/determine the combine effect of temperature and magnetic fields as they have opposite effect. However, in my opinion at high temperature the magnetic field is reduce so in such as case I think the materials will follow the effect of temperature.

Comment 10: Doping engineering, what is the effect of this on diffusion energy?

Response: Doping of heteroatom increase the electronic cloud which ultimately improve electrochemical application. Secondly, doping of heteroatom increase the adsorption of alkali metal ions, which also give favor for batteries application. A theoretical (DFT) study [1] mentioned that doping of heteroatom also enhance diffusion kinetics and this may be due to interlayer space or wide pore in materials due to doping of heteroatoms.

Reference

Sun, Xiaoli, and Zhiguo Wang. "Adsorption and diffusion of lithium on heteroatom-doped monolayer molybdenum disulfide." Applied Surface Science 455 (2018): 911-918.

Comment 11: Check the subscripts.

Response: Thank you to mention, the manuscript is modified and correct all subscripts accordingly.

Reviewer 3 Report

Sayyar Ali Shah et al reported “Recent Advancement and Structural Engineering in Transition Metal Dichalcogenides for Alkaline Metal Ions Batteries”. After careful evaluation of the manuscript, I recommend major revisions prior to publication in the Journal of “Materials”.

Comments to authors

1.     The author should discuss the alkaline metal ion batteries related materials in introduction section.

2.     The author should include synthetic diagram related changing of the morphology.

3.      The author should compare Transition Metal Dichalcogenides related with other materials.

4.      The novelty of the review work should be discussed in the introduction section of the revised manuscript.

5.      What are the main reasons for choosing Transition Metal Dichalcogenides used for alkaline metal ion batteries?

6.      Some recent signs of progress of supercapacitors shall be included in the introduction (Ionics 28, 859–869 (2022); Ionics 26, 3543–3554 (2020); Ionics 26, 5757–5772, 2020; J Mater Sci: Mater Electron 33, 8426–8434 (2022); Journal of Alloys and Compounds, 882, 15, 2021, 160409. These papers should be cited and compared in the introduction.

7.       All abbreviations in the manuscript should be given full names when they first appear.

Author Response

Dear Editor/ Reviewer

We are highly thankful to the reviewer for showing significant interest in our research work. We appreciate the reviewer’s valuable contributions in providing best possible guidance and recommendations. The highlighted incorrect statements have been corrected as red colored text enclosed in quoted marks. All the mentioned points have been corrected with best alternatives and literature agreement. Our responses to reviewers’ comments are listed as follows.

Comment 1: The author should discuss the alkaline metal ion batteries related materials in introduction section.

Response: Thank you for your suggestion, it was a mistake to mention alkaline instead of alkali but now it is corrected.

Statement in manuscript:

Recent Advancement and Structural Engineering in Transition Metal Dichalcogenides for Alkali Metal Ions Batteries

Comment 2: The author should include synthetic diagram related changing of the morphology.

Response: Thank you for your suggestion, two figures (Fig. 2 and Fig. 4) included which shows changes in morphology and surface as reviewer suggested.

Comment 3: The author should compare Transition Metal Dichalcogenides related with other materials.

Response: Thank you so much for your suggestion, but we are sorry to say that our target is chalcogenides and its surface engineering for AMIBs, so if we put a comparison or other such materials in the manuscript, the study direction will change.

Comment 4: The novelty of the review work should be discussed in the introduction section of the revised manuscript.

Response: Thank you for your suggestion, up to our knowledge the main novelty of this review is combine advancement in L-TMDs, its applications, and surface engineering. Previously, surface engineering was separately combine in review and application is separate here we combine both of the topics.

Statement in manuscript:

In this review, synthesis, modification and its application for sodium and lithium ion batteries has been summarized. In first part, synthetic approaches for development of 0D, 1D, 2D and 3D morphology based TMD electrocatalyst for AIMBs is discussed. In the second part, intrinsic modification and/or structural engineering which can further improve batteries application of TMDs has been discussed. And in the last part, the electrochemical applications of lithium and sodium ion batteries are discussed.

Comment 5: What are the main reasons for choosing Transition Metal Dichalcogenides used for alkaline metal ion batteries?

Response: Thank you for your suggestion. TMDs are emerging materials for energy applications especially for batteries. And secondly, TMDs are discussed for synthesis and application separately and surface engineering separately, so here we combine these two areas of topic. 

Comment 6: Some recent signs of progress of supercapacitors shall be included in the introduction (Ionics 28, 859–869 (2022); Ionics 26, 3543–3554 (2020); Ionics 26, 5757–5772, 2020; J Mater Sci: Mater Electron 33, 8426–8434 (2022); Journal of Alloys and Compounds, 882, 15, 2021, 160409. These papers should be cited and compared in the introduction.

Response: Authors are thankful for that comment, but unfortunately we cannot agree as the materials discussed in our manuscript rely to chalcogenides and not oxides as referee suggested. Also the purpose of this review is to present progress in batteries not the supercapacitors, being a different field of electrochemical studies.

Comment 7: All abbreviations in the manuscript should be given full names when they first appear.

Response: Thank you for your suggestion, the manuscript has been modified according to suggestion.

Round 2

Reviewer 3 Report

The authors took into account the recommendations made and considerably improved the work. The current form of the manuscript is acceptable.